# A study on substitution characteristics and competitive performance of Chinese super league teams under the five-substitution rule

Tong Chen[◉], Liang Chen[ID]‡*

School of Physical Education And Sport Science, Fujian Normal University, Fuzhou, Fujian, China

◉ This author contributed to Original Draft Preparation, editing, data collection and Methodology
‡ This author contributed to Supervision and Review
* cullencl@126.com

## Abstract

Under the revised substitution regulations in soccer, this study aims to analyze the performance characteristics of substitution events in Chinese Football Association Super League (CSL) matches and explore the relationship between soccer substitution networks and team performance over the season. Using non-goalkeeper substitution events (n=2125) from the 2023 CSL season as the research sample, the study conducted substitution network modeling and clustering to investigate the relationship between network characteristics and team performance, as well as the performance traits of different types of substitution events. Results showed that professional soccer teams with higher substitution network density demonstrated better performance in goal difference and goals conceded. Out-degree centralization positively influenced goal difference, goals scored, and team ranking, while higher In-degree centralization correlated with fewer goals conceded. Additionally, players with more balanced on-field participation were substituted most frequently. Offensive players with strong attacking abilities were substituted in earlier, while wide players were the most frequently substituted out. Players with a high frequency of high-risk passes were substituted out later. Therefore, a substitution strategy centered around some soccer players contributes to better seasonal performance. Substitutions out of players are more focused on physical condition, with high-performing teams tending to substitute out similar types of players at different times. Substitutions in, on the other hand, emphasize the compatibility of player abilities. Under the "five substitutions" rule, greater attention is given to the running distance and maximum sprint speed of soccer substitute players.

**Data availability statement:** Data cannot be shared publicly because of copyright. Data are available from the OPTA Institutional Data Access for researchers who meet the criteria for access to confidential data. The data underlying the results presented in the study are available from 17710663879@163.com

**Funding:** The National Social Science Fund of China [23BTY044] was awarded to Ph.D. Liang Chen, who is the corresponding author of this manuscript. The Fujian Province Social Science Planning Youth Project [FJ2022C019] was awarded to Ph.D. Zhe Jiang, who is not listed as an author of this manuscript. The funders had no role in the study design, data collection, data analysis, decision to publish, or manuscript preparation. However, the National Social Science Fund of China [23BTY044] primarily supported the theoretical framework development and statistical analysis of this study. The Fujian Province Social Science Planning Youth Project [FJ2022C019] provided financial support for research-related expenses, such as data processing and technical resources.

**Competing interests:** The authors have declared that no competing interests exist.

## Introduction

Due to the COVID-19 pandemic, FIFA began trialing a new rule allowing five substitutions per match in 2020. In 2022, the International Football Association Board (IFAB) officially announced the permanent adoption of this rule, increasing the number of substitute players from 12 to 15. This marked a formal update to substitution regulations in professional football matches. In professional football, team performance relies on both players and coaches. Coaches use substitutions to manage fatigue and improve performance during matches and throughout the season [1,2]. As match schedules become more congested, players experience higher physical loads, which increases their risk of injury [3]. The new rule's allowance for additional substitutions can substantially reduce player load (by 46%) [2], making it especially crucial for fatigue management.

The impact of substitutions on player and match performance has drawn significant academic interest. Studies have linked goals and victories to the timing of substitutions [4–6] and shown that the nature of the substitutions can affect team tactics and match intensity [7,8]. However, several critical issues remain unresolved. First, current research does not integrate match performance with the reasons behind substitution decisions, leaving coaches' objectives for substitutions largely unknown. Second, analyses based solely on match outcomes fail to capture the nuances of substitution strategies and lineup management. Specifically, as teams aim for season-long success rather than just single-match victories, the relationship between substitutions and match outcomes does not adequately reflect the long-term patterns and effectiveness of substitution strategies. Thus, there is a need to model teams' overall substitution characteristics across a season to explore their impact on season performance.

To uncover long-term substitution patterns, this study introduces Social Network Analysis (SNA) [9] to quantitatively model team substitution events over a season. Substituted players and their connections are represented as nodes and links, respectively. SNA is widely used in football passing analysis and effectively captures player interactions [10]. SNA can capture the inherent network structure of player interactions, offering a global perspective, such as revealing inherent playing styles [11] or identifying key players [12]. Various network metrics, such as clustering coefficient (CC) and centrality [13], can quantitatively describe network features. Similarly, substitution networks in football can be generated from substitution events between players, with network metrics calculated to quantify substitution characteristics. This study aims to analyze the relationship between team season performance and substitution events based on substitution networks and player types while exploring the performance characteristics of substituted players.

## Methods

### Sample and data

The study focuses on 2,125 substitution events (excluding goalkeepers) from 240 soccer matches involving 16 teams during the 2023 Chinese Football Association

Super League (CSL) season from April 15th to November 4th, excluding invalid samples (17 substitute players, 77 substituted-out). The dataset includes 480 substitution matrices derived from OPTA's physical and technical reports, alongside running and technical metrics of substituted players. Data were sourced from CSL's official data provider, OPTA. All the statistics were calculated as per-minute form at season level.

## Network metrics

In this study, we define the substitution network as a social network constructed based on the substitution patterns of a team during matches. The nodes in the network represent players, while edges between nodes indicate substitution relationships, where a directed link is formed from the substituted player to the substitute who replaces them. By aggregating substitution data over the course of a season, we construct a comprehensive substitution network. Five substitution network metrics were extracted to understand collective substitution characteristics, including: Network Density (ND), Clustering Coefficient (CC), Out-Degree Centralization of starters(ODC), In-Degree Centralization of non-starters(IDC).

ND measures the interconnectedness of players, defined as the ratio of existing links between substituted and replaced players to the total possible links [14]. Higher network density indicates that the team employs a greater variety of substitution combinations. This suggests that the coach utilizes a wider range of players when making tactical adjustments. Conversely, a lower network density implies fewer substitution combinations, indicating a more fixed substitution strategy that relies on a specific set of substitute players.

$$D = m / (n(n-1))$$

Where m is the actual number of substitution connections, and n is the number of nodes (players).

CC reflects the degree to which nodes (players) cluster together. The local clustering coefficient $C_i$ measures the interconnectivity among a node's neighbors, and the overall clustering coefficient is the average of local values, representing the network's clustering level [15]. Higher values indicate closer substitution relationships among local members.

$$C_i = \frac{|\{a_{jk}, a_{jk} \in E\}|}{k_i(k_i-1)}$$

$$\overline{C} = \frac{1}{n} \sum_{i=1}^{n} C_i$$

Degree Centralization (DC) Quantifies the number of direct links a node has in the network. For directed networks, it includes ODC and IDC. DC reflects the overall integration or consistency of the team, with higher values indicating greater differences in degree centralization and a stronger trend of centralization in the substitution network.

$$C_D = \frac{\sum_{i=1}^{n} [C_{max} - C_D(p_i)]}{max\{\sum_{i=1}^{n} [C_{max} - C_D(p_i)]\}}$$

## Statistical analysis

Pearson Correlation Analysis used to examine relationships between substitution network metrics, team seasonal performance, player category proportions, and team rankings. Correlation coefficients (r) were categorized as follows: |0~0.29|: very weak, |0.3~0.49|: low, |0.5~0.79|: moderate, | > 0.8|: high correlation.

Players were categorized by K-Means Clustering Analysis based on substitution frequency (in/out) and average playing time. Based on the cluster ing result, we adopted the Multinomial Logistic Regression to explore how running and technical performance metrics jointly influenced substitution events.

After standardizing variables using Z-scores, univariate ANOVA and multicollinearity diagnostics were performed to filter independent variables. Variables meeting P< 0.10 and VIF<5 criteria were included in the regression model. To further assess the impact of each factor on substitution events, the Odds Ratio (OR) was incorporated. OR values represent multiplicative effects, calculated by exponentiating the regression coefficient (OR=$e^{\beta}$). An OR>1 indicates a favorable factor, OR<1 indicates an unfavorable factor, and OR=1 suggests no relationship between the variable and the substitution event. Finally, the Spearman' s correlation tests were carried out to explore how the components of the teams' rosters influenced teams' performance. All the statistical procedures were conducted on Ucinet 7.0 and IBM SPSS Statistics for Windows Version 27.0. Armonk, NY: IBM Corp with a statistical significance of 0.05.

## Results

### The correlation between substitution network metrics and the teams' season performance

The Pearson correlation analysis (Table 1) revealed no significant relationships between the CC of local substitution networks and season performance (P>0.05). However, overall network characteristics showed partial correlations: ND was positively correlated with goal difference (r=0.558) and negatively with goals conceded (r=-0.669; P < 0.05) but had no significant link to rankings or goals scored (P>0.05). ODC was positively correlated with goal difference (r=0.499) and goals scored (r=0.515) and negatively with rankings (r=−0.547; P<0.05), while no correlation with goals conceded was found (P>0.05). IDC was negatively correlated with goals conceded (r=−0.512; P<0.05) but showed no significant correlation with rankings, goals scored, or goal difference (P>0.05).

### Modelling substitution and replaced players'variables for obtained clusters

Substitution frequency and average playing time were used as the input variables of the K-Means clustering. Three types of substitute players and replaced players were identified (Fig 1). substitute players players: Type 1 with the fewest average appearances (2.98) and the shortest playing time (18.39 minutes). Type 2 with moderate average playing time (39.91 minutes) and slightly higher appearances (3.57). Type 3 with the highest average appearances (9.61) but moderate single-game playing time (27.21 minutes).

Replaced players: Type 1 with shorter average playing time (52.36 minutes) and fewer substitutions (2.89). Type 2 with the longest average playing time (74.92 minutes) and moderate substitutions (4.36). Type 3 with long average playing time (70.52 minutes) and the highest number of substitutions (11.20).

### The dominant factor in the type of substitution and replaced playes

To investigate the relationship between technical and running performance and player classifications, multinomial logistic regression was conducted. Substitution types were dependent variables, and performance metrics served as independent variables. After filtering variables via univariate ANOVA (P<0.10) and multicollinearity diagnostics (VIF<5) key indicators were identified (Table 2).

**Table 1. Correlation analysis between substitution network characteristics and seasonal performance in CSL teams.**

|  | ND | CC | ODC | IDC |
|---|---|---|---|---|
| Standing | −0.345 | 0.026 | −0.547* | −0.107 |
| Goals | 0.342 | −0.109 | 0.515* | 0.207 |
| Goals conceded | −0.669** | −0.18 | −0.406 | −0.512* |
| Goal difference | 0.558* | 0.046 | 0.499* | 0.399 |

**=P < 0.01;

*=P < 0.05.

**(a) Substitute Player**

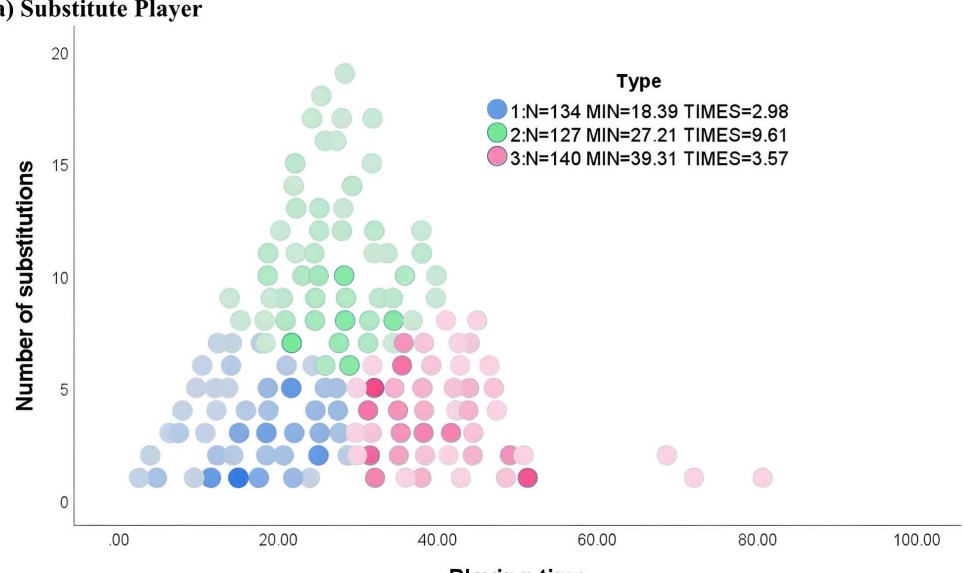

**(b) Replaced Player**

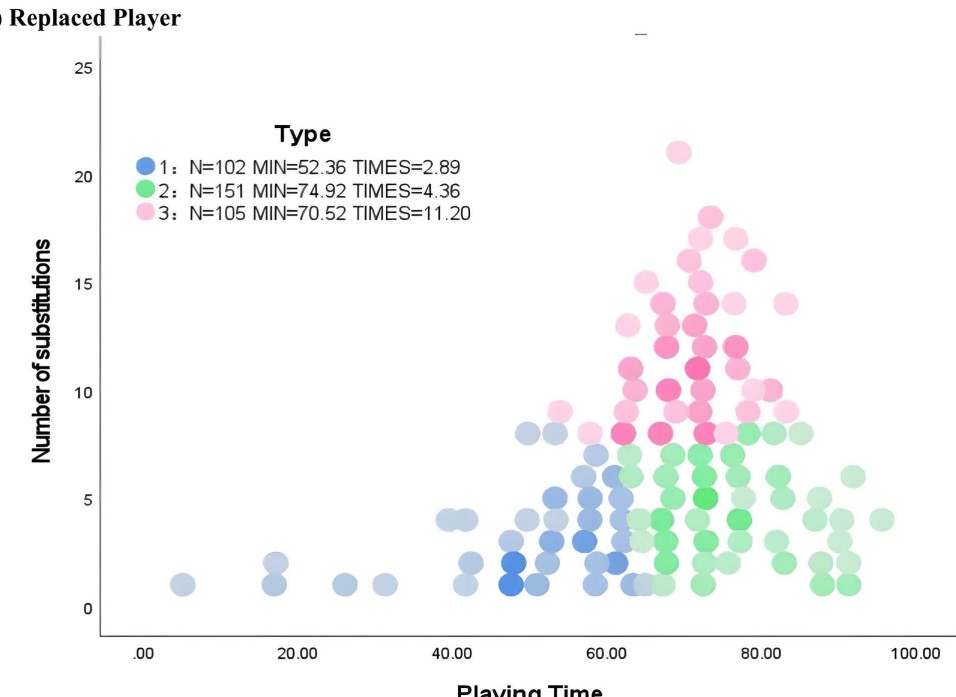

**Fig 1.  The illustration of clustering substitute players and replaced players.**

## The dominant factor in the type of substitution playes

Among the substitution playes, Logistic regression results (Table 3) showed that compared to Type 1, Type 2 and Type 3 players had lower passing accuracy (OR=0.247, r=0.866), greater jogging distance (OR=1.487, r=1.270), and slower maximum speed (OR=0.028, 0.778), with Type 2 also making more passes (OR= r=1.345) and Type 3 achieving more

**Table 2. One-way ANOVA and multicollinearity diagnostic results.**

| Variables | | Substitution | | | Replaced | | |
|---|---|---|---|---|---|---|---|
| | | F | p | VIF | F | p | VIF |
| Technical | Personal ball control | 1.412 | 0.244 | | 1.283 | 0.277 | – |
| | Passing | 2.939 | 0.053 | 1.175 | 2.419 | 0.089 | 2.393 |
| | Passing accuracy | 43.496 | 0 | 1.363 | 68.597 | 0 | 1.763 |
| | Forward passes | 2.077 | 0.126 | – | 18.414 | 0 | 2.729 |
| | Forward pass accuracy | 9.692 | 0 | 1.429 | 25.382 | 0 | 1.774 |
| | Possession loss | 1.378 | 0.252 | – | 0.036 | 0.965 | – |
| | Active crosses | 0.867 | 0.42 | | 11.837 | 0 | 1.109 |
| | Successful crosses | 3.18 | 0.042 | 1.007 | 1.233 | 0.292 | – |
| | Passes into the attacking third | 1.135 | 0.322 | – | 1.163 | 0.313 | – |
| | Key passes | 1.305 | 0.271 | – | 16.217 | 0 | 1.121 |
| | Shots | 2.091 | 0.124 | – | 0.597 | 0.55 | – |
| | Shots on target | 0.495 | 0.61 | – | 1.733 | 0.177 | – |
| | Goals | 0.147 | 0.863 | – | 0.369 | 0.691 | – |
| | Gained possession | 3.474 | 0.031 | 1.226 | 13.804 | 0 | 1.481 |
| | Tackles | 2.62 | 0.073 | 1.05 | 1.169 | 0.311 | – |
| | Successful tackles | 0.456 | 0.634 | – | 1.33 | 0.265 | – |
| | Tackle success rate | 1.581 | 0.206 | – | 0.984 | 0.374 | – |
| | Aerial Challenge for the ball | 2.212 | 0.11 | – | 18.834 | 0 | 10.239 |
| | Successful aerial Challenge for the ball | 0.232 | 0.793 | – | 32.67 | 0 | 13.508 |
| | Aerial Challenge for the ball success rate | – | – | – | 8.095 | 0 | 4.402 |
| | Ground Challenge for the ball | 2.648 | 0.071 | 1.019 | 0.394 | 0.674 | – |
| | Successful ground Challenge for the ball | 0.835 | 0.434 | – | 0.004 | 0.996 | – |
| | Ground Challenge for the ball success rate | 0.583 | 0.558 | – | 0.101 | 0.904 | – |
| | Challenge for the ball | 2.217 | 0.109 | – | 11.28 | 0 | 10.36 |
| | Total successful Challenge for the ball | 0.037 | 0.964 | – | 23.531 | 0 | 13.654 |
| | Challenge for the ball success rate | 1.794 | 0.167 | – | 9.24 | 0 | 4.417 |
| Physical | Total distance | 20.375 | 0 | 5.946 | 6.996 | 0.001 | – |
| | Standing distance | 7.314 | 0.001 | 1.601 | 9.306 | 0 | 1.154 |
| | Jogging distance | 25.792 | 0 | 1.37 | 4.734 | 0.009 | 1.209 |
| | Low speed running distance | 24.4 | 0 | – | 2.375 | 0.093 | 2.032 |
| | Medium speed running distance | 19.906 | 0 | 4.926 | 14.346 | 0 | 3.573 |
| | High speed running distance | 17.842 | 0 | 7.915 | 6.812 | 0.001 | 11.758 |
| | Sprint distance | 6.401 | 0.002 | 4.88 | 4.601 | 0.01 | 8.051 |
| | High speed running count | 14.571 | 0 | 5.85 | 7.061 | 0.001 | 9.514 |
| | Sprint count | 6.778 | 0.001 | 4.645 | 5.272 | 0.005 | 7.728 |
| | Maximum speed | 22.913 | 0 | 1.433 | 95.752 | 0 | 1.054 |

Standing=0~0.2 m/s; Jogging=0.2~2 m/s; Low speed running=2~4 m/s; Medium speed running=4~5.5 m/s; High speed running=5.5~7 m/s; Sprint= > 7 m/s.

**Table 3. The impact of performance on the type of substitute players base on Type 1.**

| Variables | | Type2 | | | | Type3 | | | |
|---|---|---|---|---|---|---|---|---|---|
| | | R | Wald | p | OR | R | Wald | p | OR |
| Technical | Passing | 0.297 | 9.269 | 0.002 | 1.345 | −0.006 | 0.006 | 0.936 | 0.994 |
| | Passing accuracy | −1.398 | 97.951 | 0 | 0.247 | −0.144 | 7.839 | 0.005 | 0.866 |
| | Forward pass accuracy | −0.111 | 1.101 | 0.294 | 0.895 | −0.045 | 0.348 | 0.555 | 0.956 |
| | Successful crosses | 0.173 | 3.77 | 0.052 | 1.189 | 0.179 | 4.855 | 0.028 | 1.196 |
| | Gained possession | −0.04 | 0.206 | 0.65 | 0.961 | −0.067 | 1.024 | 0.312 | 0.936 |
| | Tackles | −0.122 | 2.568 | 0.109 | 0.885 | −0.087 | 2.432 | 0.119 | 0.917 |
| | Ground Challenge for the ball | −0.02 | 0.059 | 0.808 | 0.98 | 0.12 | 3.51 | 0.061 | 1.127 |
| Physical | Standing distance | 0.114 | 0.936 | 0.333 | 1.121 | 0.017 | 0.027 | 0.871 | 1.017 |
| | Jogging distance | 0.397 | 10.162 | 0.001 | 1.487 | 0.239 | 4.277 | 0.039 | 1.27 |
| | Low speed running distance | −0.146 | 1.164 | 0.281 | 0.864 | −0.015 | 0.017 | 0.897 | 0.985 |
| | Medium speed running distance | 0.113 | 0.772 | 0.38 | 1.12 | 0.115 | 1.584 | 0.208 | 1.121 |
| | Sprint distance | 0.218 | 1.182 | 0.277 | 1.244 | 0.1 | 0.659 | 0.417 | 1.105 |
| | Sprint count | −0.103 | 0.289 | 0.591 | 0.902 | 0.006 | 0.002 | 0.962 | 1.006 |
| | Maximum speed | −3.581 | 81.751 | 0 | 0.028 | −0.251 | 6.013 | 0.014 | 0.778 |

successful crosses (OR=1.196). Defensive metrics and high-intensity running indicators did not significantly differentiate Type 1 from the others.

### The dominant factor in the type of replaced players

Table 4, With Type 1 and Type 2 as the reference, revealed that Type 2 players were more likely to have lower passing accuracy (OR=0.502), less jogging distance (OR=0.298), fewer medium-speed runs (OR=0.298), and slower maximum speed (OR=0.315); Type 3 players had more passes (OR=1.411), lower passing accuracy (OR=0.603), fewer forward passes (OR=0.667), more crosses (OR=1.215), more key passes (OR=1.185), more standing distance (OR=1.194), fewer medium-speed runs (OR=0.66), more sprinting distance (OR=1.800), and slower maximum speed (OR=0.519). Defensive metrics, such as ball recovery and aerial duels, did not significantly affect the classification of Type 2 or Type 3 substitute players (p > 0.05). When comparing Types 2 and 3, Type 3 players exhibited fewer forward passes (OR=0.666), more crosses (OR=1.195), more key passes (OR=1.187), faster maximum speed (OR=1.373), more jogging distance (OR=3.650), and more sprinting distance (OR=2.051).

### The correlation between the teams' season performance and the type of substitute players

The results of the correlation analysis only showed a correlation between the proportion of substitutions and the team's season performance (Table 5). A higher proportion of Type 3 substitute players was associated with better team ranking (r=−0.603*), more goals (r=0.684**), fewer goals conceded (r=−0.519*), and greater goal difference (r=0.651**).

## Discussion

This study aimed to explore the relationship between soccer rotation strategies and season-level competitive performance under the five-substitution rule. The results revealed moderate correlations between ND, IDC, ODC, and key performance metrics, including team rankings, goals scored, goals conceded, and goal difference. Three types of substitute and replaced players were identified, and their performance differences were examined. Additionally, teams with a higher

**Table 4. The impact of performance on the type of replaced players based on Type 1.**

| Variables | | Type 2 | | | | Type 3 | | | |
|---|---|---|---|---|---|---|---|---|---|
| | | R | Wald | p | OR | R | Wald | p | OR |
| Technical | Passing | 0.194 | 2.618 | 0.106 | 1.214 | 0.345 | 9.711 | 0.002 | 1.411 |
| | Passing accuracy | −0.69 | 32.293 | 0 | 0.502 | −0.505 | 29.04 | 0 | 0.603 |
| | Forward passes | −0.023 | 0.036 | 0.849 | 0.977 | −0.405 | 12.424 | 0 | 0.667 |
| | Forward pass accuracy | −0.082 | 0.548 | 0.459 | 0.921 | 0.024 | 0.072 | 0.789 | 1.025 |
| | Active crosses | 0.024 | 0.07 | 0.792 | 1.024 | 0.195 | 5.828 | 0.016 | 1.215 |
| | Key passes | −0.002 | 0.001 | 0.98 | 0.998 | 0.17 | 4.24 | 0.039 | 1.185 |
| | Gained possession | 0.043 | 0.225 | 0.635 | 1.043 | −0.032 | 0.146 | 0.702 | 0.968 |
| | Aerial Challenge for the ball success rate | 0.01 | 0.007 | 0.933 | 1.01 | −0.037 | 0.117 | 0.733 | 0.963 |
| | Challenge for the ball success rate | 0.041 | 0.116 | 0.734 | 1.042 | −0.094 | 0.727 | 0.394 | 0.911 |
| Physical | Standing distance | 0.011 | 0.007 | 0.934 | 1.011 | 0.177 | 5.069 | 0.024 | 1.194 |
| | Jogging distance | −1.212 | 32.333 | 0 | 0.298 | −0.063 | 0.56 | 0.454 | 0.939 |
| | Low speed running distance | −0.562 | 11.444 | 0.001 | 0.57 | −0.416 | 16.445 | 0 | 0.66 |
| | Medium speed running distance | −0.089 | 0.583 | 0.445 | 0.915 | 0.588 | 31.512 | 0 | 1.8 |
| | Maximum speed | −1.046 | 91.259 | 0 | 0.351 | −0.657 | 67.08 | 0 | 0.519 |

**Table 5. The 95% confidence intervals of correlations between teams' season performance and percentages of three clusters of substitute players.**

| Teams' season performance | Percentage of type 1 | Percentage of type 2 | Percentage of type 3 |
|---|---|---|---|
| Standing | 0.366 | 0.302 | −0.603* |
| goals | −0.41 | −0.344 | 0.684** |
| goals conceded | 0.436 | 0.172 | −0.519* |
| goal difference | −0.461 | −0.277 | 0.651** |

proportion of Type 3 substituted players exhibited better seasonal performance.This research represents the first attempt to apply social network analysis (SNA) to the study of football substitutions, offering a quantitative perspective on understanding overall rotation structures. The findings emphasize the link between team performance and long-term rotational patterns while highlighting variations in player-type distributions across teams. These insights can assist coaches and managers in understanding the dynamics of rotation and optimizing their squad configurations.

### The relationships between substitution network metrics and team performance

A significant correlation was found between the substitution ND of CSL teams and both goal difference (r=0.558) and goals conceded (r=−0.669). Substitution ND represents the ratio of existing substitution links between substituted and replaced players to the maximum possible links. Teams with higher substitution ND exhibit stronger connections among individual players within the overall substitution events, indicating greater player variability and more flexible tactical changes. In football, substitutions can be categorized as positional (like-for-like) or non-positional. Positional substitutions prioritize physical and technical attributes, while non-positional substitutions focus on tactical adjustments. Teams with higher substitution ND are more likely to employ non-positional substitutions, suggesting a broader range of tactical

adjustments. Research has demonstrated that tactical changes in formation can directly impact team performance, particularly by creating more goal scoring opportunities after such adjustments [16]. Although the number of substitutions allowed per match has increased, the utilization rate of these substitutions by teams has generally decreased [17]. This trend may indicate that not all teams possess sufficient bench depth ("deep benches") and also highlights the potential advantage of a more diverse substitution network.

At the player level, this study found that greater centrality of both replaced and substitute players within the substitution network in the substitution network correlates with better team performance over the season. The centrality of replaced players was positively correlated with goal difference (r=0.499) and goals scored (r=0.515), and teams with more stable replaced players were more likely to achieve greater seasonal rankings (r=-0.547). For substitute players, their centrality was significantly negatively correlated with goals conceded (r=−0.512). Regarding match performance [18], found that substitute players accounted for only 10.76% of goals scored, indicating that replaced or entire match players participated in more goal-scoring opportunities. Additionally, multiple studies have highlighted the context-specific relationship between substitutions and goal outcomes, considering factors such as home vs. away matches [19,20] and scoreline status [6,20]. A high degree of substitute or replaced player centrality reflects relatively consistent substitution arrangements for players, which seem to play a critical role in maintaining the team's offensive and defensive balance. Core players are vital to a team's passing network and tactical systems. Since players' physical performance typically declines significantly in the final 20 minutes of a match [21], a drop in core player performance can negatively impact overall team performance. Thus, stable rotations and capable substitute players for core players appear necessary for consistent team performance.

The findings of this study indicate that defensive stability relies heavily on solid personnel arrangements, while team seasonal performance benefits from stable substitute players rotations. When substitution decisions align with a well-established tactical system, team cohesion and overall synergy can be better maintained, thereby enhancing seasonal performance. Therefore, maintaining a stable substitution structure for key positions or players may contribute positively to a team's seasonal success.

## Game-related statistics of each substitute players cluster on individual performance

Cluster analysis revealed significant differences in the playing time and substitution frequency characteristics of players in different substitution types, potentially reflecting CSL coaches' strategies for choosing substitute or replaced players during matches. The average number of appearances for the three substitute player types was 2.98, 3.57, and 9.61, corresponding to playing times of 18.39 minutes, 39.91 minutes, and 27.21 minutes, respectively. This study identified that the primary distinctions among substitute player categories in CSL teams were observed in passing (frequency and accuracy), cross passing, jogging distance, and maximum speed.

From a technical perspective, previous research has confirmed the association of passing and cross passing with team rankings over a season [22]. However, studies on substitution-related technical statistics are limited to comparing substitute, replaced, or entire match players [23,24]. Under the "three-substitution" rule, studies has shown that substitute players' high-intensity running and maximum speed are influenced by their playing time. For example, Liu [25] observed that late substitutes (<22.5 minutes) covered greater distances in medium-to-high intensity running (>15 km/h) than early substitutes (≥22.5 minutes) but exhibited lower maximum speeds and reduced jogging performance. Similarly, Hills [26] reported that substitute players introduced in the last 15 minutes of matches covered shorter high-speed distances and had lower average accelerations than those substitute players between the 60th and 75th minutes.

Under the "five-substitution" rule, this study considered not only substitution timing but also substitution frequency. Results revealed significant differences in jogging distance and maximum speed, whereas medium-to-high intensity running showed no significant variation. Although high-intensity running is a valid indicator of physical performance in soccer, it does not necessarily reflect a player's overall contribution [27]. The findings regarding maximum speed are consistent with previous studies, likely because maximum speed in soccer often occurs in key moments, such as gaining possession

or surpassing an opponent [28]. Additionally, substitution timing has been shown to correlate with goal occurrences and match outcomes [4,18,20]. Under the "five-substitution" rule, jogging distance and maximum speed may be the key factors influenced by substitution frequency and playing time rather than medium-to-high intensity running.

Previous studies have demonstrated that substitute players can enhance team performance, and their contributions to goals are a crucial factor for championship success [29]. This study further found that type 3 substitutes, who were used most frequently, were positively associated with a team's seasonal performance; that is, the more concentrated the substitutions, the better the team's overall performance, also corroborating social network analysis results. When making substitution decisions, coaches tend to prefer balanced substitute players (63.65%) [30]. Similarly, this study showed that CSL teams most frequently used Type 3 substitute players, who were characterized by fewer passes, higher passing accuracy, more ground duels, moderate jogging distances, and maximum speeds. While passing frequency alone does not impact match outcomes [31], passing accuracy plays a critical role in maintaining possession, creating scoring opportunities, and minimizing opponents' possession time [32]. In contrast, Type 1 substitute players had the highest passing accuracy but the lowest playing time and substitution frequency, potentially indicating that these players are often starters or that CSL teams lack sufficient reserves of such players. Consequently, relatively balanced performance substitute players appear more practical, with a greater proportion contributing positively to team performance over the season.

Additionally, Type 1 substitute players exhibited low passing frequency, limited successful crosses, the shortest jogging distances, and the highest maximum speeds. This combination of attributes suggests that Type 1 substitute players are more focused on quick progression or distribution following ball possession gain rather than sustained possession or positional interplay. Studies have shown that substitutions tend to occur later when facing stronger opponents [33]. Thus, Type 1 substitute players are more likely to be used in the final stages of matches against stronger opponents to enhance counterattack efficiency through high-accuracy passing and rapid transitions. Type 2 substitute players exhibited the highest playing time and passing frequency but the lowest passing accuracy, engaged less in ground duels, covered the greatest jogging distances, and had the lowest maximum speeds. This indicates that early substitute players often assume high-risk passing roles or responsibilities in offensive organization and progression, potentially as key links in midfield or defensive-to-offensive transitions. However, in CSL matches, teams that lost or drew matches tended to have higher passing volumes but lower passing accuracy, indicating that efficient offensive organization depends more on passing accuracy rather than frequency [34]. The lower passing accuracy of Type 2 substitute players may lead coaches to use them more cautiously.

### Game-related statistics of each replaced players cluster on individual performance

Existing research on replaced players has primarily focused on comparisons with substitute players and entire-match players. In soccer,Substitutions are typically made to reduce team fatigue or adjust tactics [35,36]. To achieve these objectives, substitute players are expected to perform at least as well as starters [36,37]. Current studies indicate that substitutes generally exhibit superior technical and average running performance but achieve lower peak running distances and reduced high-intensity running within 1-, 2-, and 5-minute time intervals, with positional differences also observed [23,24,38]. This study is the first to conduct intra-group comparisons of replaced players. Although the substitution network analysis revealed that teams with higher replaced players centrality performed better over the season, the correlation analysis after clustering showed no significant relationship between replaced player types and team season performance. This discrepancy may be attributed to the inclusion of playing time as a factor in clustering, which also suggests that high-performing CSL teams frequently replace players of similar types at different times. This pattern likely results from substitution timing being influenced by factors such as scoreline, ranking gaps, extra time, match phases, and substitution rule [30].

Replaced players were categorized into three types in this study, with average substitution frequencies of 2.89, 4.36, and 11.20 times, corresponding to playing times of 52.36 minutes, 74.92 minutes, and 70.52 minutes, respectively. On one hand, this aligns with the timing and frequency of substitute players, as core players in the substitution network are more likely to substitute or be replaced during the middle of the second half. On the other hand, compared to substitute

players, replaced players exhibited greater differences in running performance. This differences arises because coaches prioritize replaced player's physical condition when making substituting, whereas decision regarding substitute players focus more on physical quality, such as maximum speed, rather than physical fitness.

Type 1 replaced players were characterized by lower passing and cross passing frequency but higher passing accuracy, more forward and key passes, and superior running performance, including the highest moderate-speed running distance and maximum speed. Given their lower substitution frequency and shorter average playing time, these players are likely core offensive players who are rarely replaced. In contrast, both Type 2 and Type 3 replaced players had average playing times exceeding 70 minutes, a stage where players typically experience significant declines in physical performance [21]. Therefore, their substitutions were likely influenced by fatigue-related considerations.Type 2 replaced players were characterized by more forward passes, lower passing accuracy, and the weakest running performance,including the shortest running distances and slowest maximum speed. Conversely, Type 3 replaced players exhibited fewer forward passes but more live-ball crosses and key passes, along with better performance in maximum speed, jogging, and high-speed running distances. These findings suggest that Type 2 replaced players were likely midfield or attacking players who attempted frequent forward passes but had lower success rates. Type 3 replaced players, in contrast, were likely highly substitutable wide players, whose high-intensity sprints and accelerations in the latter stages of the second half may have contributed to their fatigue [39], explaining their frequent substitutions.

Despite providing valuable insights into substitution strategies in CSL teams, this study has several limitations. First, the analysis is based on data from a single season, which may limit the generalizability of the findings. Second, this study does not fully account for goal difference and match context, all of which can significantly influence substitution decisions. Thus, further research is needed to examine how substitution strategies interact with goal difference, match phase, and opposition strength, while multi-season and cross-league analyses, along with player workload metrics, could enhance understanding and predictive accuracy of substitution decisions.

These findings may be useful for coaches when they deciding substitute or replaced players. Coaches can optimize their substitution strategy by considering players' importance within the substitution network and their performance. A highly centralized substitution network may contribute to greater team seasonal performance.Therefore, maintaining consistent substitute or replaced players in critical positions to ensure that they can seamlessly integrate into the tactical system could be beneficial.

## Conclusions

In conclusion, this study applied a Social Network Analysis (SNA) framework to generate substitution networks and extract rotation characteristics, exploring the relationship between seasonal performance and long-term rotation strategies. ND, ODC, and IDC were moderately correlated with team rankings, goals scored, goals conceded, and goal differentials, indicating that a stable substitution structure centered around core players contributes to better seasonal performance. Based on players' rotation frequency and playing time, three distinct types of substitutes and replaced players were identified. Results from the multinomial unordered logistic regression model demonstrated that substitutions of replaced players were more focused on fitness factors, with high-performing teams tending to replace similar types of players at different times. In contrast, substitutions of incoming players prioritized adaptability to the team's tactical needs, with the "five-substitution" rule placing greater emphasis on jogging distance and maximum speed for substitutes. These findings provide coaches and managers with insights into seasonal rotation characteristics of teams and players, summarize the performance attributes of substitutes and replaced players, and offer a quantitative tool for optimizing rotation strategies and roster management.

## Supporting information

The data underlying the findings of this study have been anonymized and are available in the supplementary files. These include the team substitution information of 2023 season CSL teams, including team season performance, substitution matrix, substituted and replaced players' performance. Due to the copyright restrictions imposed by the official data

provider, Opta, all personally identifiable information, including player names and club affiliations, has been anonymized. This anonymization ensures that the data are free from any intellectual property concerns while still preserving the integrity of the statistical analyses and outcomes.

**S1 Table. Correlation analysis between substitution network characteristics and seasonal performance in CSL teams.** S1 Table was primarily used to analyze the relationship between team substitution networks and team performance throughout the season, including the substitution matrices of each team.
(XLSX)

**S1 Fig. The illustration of clustering substitute players and replaced players.** S1 Fig primarily categorizes players based on their playing time and the number of substitutions, including both substituted and replaced players.
(ZIP)

**S2 Table. One-way ANOVA and multicollinearity diagnostic results.** S2 Table was primarily employed to identify suitable features for subsequent analyses, encompassing data derived from univariate variance analysis and collinearity diagnostics.
(ZIP)

**S3 Table. The impact of performance on the type of substitute players base on Type 1.** S3 Table was employed to analyze performance differences between substitute player categories, incorporating statistical evaluations of group-wise comparisons.
(XLSX)

**S4 Table. The impact of performance on the type of replaced players based on Type 1.** S4 Table was employed to analyze performance differences between replaced player categories.
(XLSX)

**S5 Table. The 95% confidence intervals of correlations between teams' season performance and percentages of three clusters of substitute players.** Table S5 was designed to investigate associations between the compositional distribution of player categories and seasonal team performance.
(XLSX)

## Author contributions

**Data curation:** Tong Chen.

**Funding acquisition:** Liang Chen.

**Methodology:** Tong Chen.

**Supervision:** Liang Chen.

**Writing – original draft:** Tong Chen.

**Writing – review & editing:** Liang Chen.

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
