## [Decision Letter · Decision Letter 0]

2 Mar 2025

PONE-D-25-03456A Study on Substitution Characteristics and Competitive Performance of Chinese Super League Teams under the Five-Substitution RulePLOS ONE

Dear Dr. Chen,

Thank you for submitting your manuscript to PLOS ONE. After careful consideration, we feel that it has merit but does not fully meet PLOS ONE’s publication criteria as it currently stands. Therefore, we invite you to submit a revised version of the manuscript that addresses the points raised during the review process.

We look forward to receiving your revised manuscript.

Kind regards,

Jovan Gardasevic

Academic Editor

PLOS ONE

Journal Requirements:

3. Thank you for stating the following financial disclosure: The work was supported by the National Social Science Foundation of China[23BYT044]

Reviewers' comments:

Reviewer's Responses to Questions

**Comments to the Author**

1. Is the manuscript technically sound, and do the data support the conclusions?

Reviewer #1: Yes

Reviewer #2: Yes

2. Has the statistical analysis been performed appropriately and rigorously? 

Reviewer #1: Yes

Reviewer #2: Yes

3. Have the authors made all data underlying the findings in their manuscript fully available?

Reviewer #1: Yes

Reviewer #2: Yes

4. Is the manuscript presented in an intelligible fashion and written in standard English?

Reviewer #1: No

Reviewer #2: Yes

5. Review Comments to the Author

Reviewer #1: It is not clear from the abstract which sport it is. That is unacceptable.

The study is interesting. It provides information that an adequate substitution strategy can improve team performance.

I would recommend that the proofreader look at this text. I think the academic style of English could be improved. There are too many complex sentences that make it difficult to understand. Like for example this one: "A stable substitution structure centered around core players is conducive to better seasonal performance." That could be said more simply. There are many such sentences throughout the text as well.

A paragraph on Limitations of the study should be added to the Discussion. Also a paragraph with recommendations for coaches. Something like this: Coaches can optimize their substitution strategy by...

I will recommend that the manuscript be published with minor corrections.

Reviewer #2: Very current and interesting topic. Especially since there is little research on this issue, and the load and recovery of athletes in modern competitive calendars can be decisive in achieving results. The results of the study are clearly presented, the methodology is respected. The language is understandable. The discussion is very detailed.

6. PLOS authors have the option to publish the peer review history of their article (what does this mean? ). If published, this will include your full peer review and any attached files.

**Do you want your identity to be public for this peer review?** For information about this choice, including consent withdrawal, please see our Privacy Policy .

Reviewer #1: **Yes: ** Bojan Masanovic

Reviewer #2: No

---

## [Author Response · Author response to Decision Letter 0]

11 Mar 2025

Dear Editor Gardasevic,

Thank you for giving us the opportunity to submit a revised draft of our manuscript titled "A Study on Substitution Characteristics and Competitive Performance of Chinese Super League Teams under the Five-Substitution Rule" [PONE-D-25-03456] to PLOS ONE. We appreciate the time and effort that you and the reviewers have dedicated to providing valuable feedback on our manuscript. We have incorporated changes that reflect all the suggestions provided by the reviewers. All changes are highlighted in red in the revised manuscript. Please see below for our point-by-point responses to the reviewers' comments. We hope that our work can be improved again.

Response to Comments of Reviewer #1

Comment 1�It is not clear from the abstract which sport it is. That is unacceptable. The study is interesting. It provides information that an adequate substitution strategy can improve team performance.

Response to comment 1:We sincerely appreciate your careful review and your interest in our research topic. Regarding your comment on the lack of clarity in distinguishing the sport in the abstract, we acknowledge that this was an unacceptable oversight on our part. We sincerely apologize for this mistake. Based on your suggestion, we have carefully revised and supplemented the abstract by explicitly specifying that our study focuses on soccer. This modification ensures that readers can clearly understand the context of our research. We genuinely hope that these improvements enhance the clarity and quality of the abstract. Thank you again for your valuable feedback.

Comment 2�I would recommend that the proofreader look at this text. I think the academic style of English could be improved. There are too many complex sentences that make it difficult to understand. Like for example this one: "A stable substitution structure centered around core players is conducive to better seasonal performance." That could be said more simply. There are many such sentences throughout the text as well.

Response to comment 2:Thank you for bringing these concerns to our attention. We sincerely apologize for any confusion caused by the previous version of the manuscript. We have taken significant efforts to enhance both the language quality and overall readability. We also sought the assistance of native English speakers to refine the language for improved precision and cohesion.

Additionally, we have conducted a thorough revision of the abstract, introduction, and discussion sections, refining inappropriate expressions and improving the overall structure and language. We have reorganized the logic and terminology to enhance the professionalism of the writing. Furthermore, we have simplified complex sentences by removing redundant elements to make the text more concise and readable. For example, we revised "A stable substitution structure centered around core players is conducive to better seasonal performance." to "a substitution strategy centered around some soccer players contributes to better seasonal performance. " (See Abstract). Similar modifications have been made throughout the manuscript to improve sentence structure and grammar. All revised sentences have been highlighted in red for your convenience in reviewing these changes.

Comment 3�A paragraph on Limitations of the study should be added to the Discussion. Also a paragraph with recommendations for coaches. Something like this: Coaches can optimize their substitution strategy by...

Response to comment 3:We are grateful for your insightful comment, which allowed us to incorporate an important discussion on the limitations of this study and practical recommendations for coaches. Acknowledging research limitations and providing guidance for soccer coaches in substitution strategies is highly meaningful, as it not only highlights areas that require further investigation but also offers practical applications for coaching decisions. We sincerely apologize for the initial omission of this crucial aspect. Based on your recommendation, we have added the following content to the discussion section:

Despite providing valuable insights into substitution strategies in CSL teams, this study has several limitations. First, the analysis is based on data from a single season, which may limit the generalizability of the findings. Second, this study does not fully account for goal difference and match context, all of which can significantly influence substitution decisions.Thus, further research is needed to examine how substitution strategies interact with goal difference, match phase, and opposition strength, while multi-season and cross-league analyses, along with player workload metrics, could enhance understanding and predictive accuracy of substitution decisions.

These findings may be useful for coaches when they deciding substitute or replaced players. Coaches can optimize their substitution strategy by considering players’ importance within the substitution network and their performance. A highly centralized substitution network may contribute to greater team seasonal performance.Therefore, maintaining consistent substitute or replaced players in critical positions to ensure that they can seamlessly integrate into the tactical system could be beneficial.(See Discussion)

Thank you once again for your constructive feedback. We truly appreciate your time and effort in reviewing our manuscript.

Response to Comments of Reviewer #2

Comment 1�Very current and interesting topic. Especially since there is little research on this issue, and the load and recovery of athletes in modern competitive calendars can be decisive in achieving results. The results of the study are clearly presented, the methodology is respected. The language is understandable. The discussion is very detailed.

Response to comment 1:We sincerely appreciate your positive feedback on our study. We are delighted that you found the topic both current and interesting, particularly given the limited research on this issue. The impact of soccer substitute within modern competitive calendars is indeed a crucial factor in the athlete load and recovery , and we are pleased that our study contributes to this important area of research.

We are also grateful for your acknowledgment of the clarity of our results presentation, the rigor of our methodology, and the comprehensiveness of our discussion. Ensuring methodological accuracy and providing a detailed interpretation of findings were key priorities in our work, and it is encouraging to receive such recognition.

Think you and best regards.

Yours sincerely

Chen Tong

Corresponding author:

Name: Chen Liang

E-mail: cullencl@126.com

---

## [Decision Letter · Decision Letter 1]

16 Mar 2025

PONE-D-25-03456R1A Study on Substitution Characteristics and Competitive Performance of Chinese Super League Teams under the Five-Substitution RulePLOS ONE

Dear Dr. Chen,

Thank you for submitting your manuscript to PLOS ONE. After careful consideration, we feel that it has merit but does not fully meet PLOS ONE’s publication criteria as it currently stands. Therefore, we invite you to submit a revised version of the manuscript that addresses the points raised during the review process.

We look forward to receiving your revised manuscript.

Kind regards,

Bruno Travassos

Academic Editor

PLOS ONE

Journal Requirements:

Additional Editor Comments:

The authors should revise the manuscript according to reviewers suggestions.

Also, before acceptance, the authors should improve the description about the concepts: substitution networks, substitution links and also a further explanation about what means in terms of the game the concept "Higher density indicates closer substitution relationships among players, while lower density reflects looser connections". The point is what means closer substitution relationship? These concepts are fundamental for a clear understanding of the manuscript and its results.

Reviewers' comments:

Reviewer's Responses to Questions

**Comments to the Author**

1. If the authors have adequately addressed your comments raised in a previous round of review and you feel that this manuscript is now acceptable for publication, you may indicate that here to bypass the “Comments to the Author” section, enter your conflict of interest statement in the “Confidential to Editor” section, and submit your "Accept" recommendation.

Reviewer #1: All comments have been addressed

Reviewer #2: (No Response)

2. Is the manuscript technically sound, and do the data support the conclusions?

Reviewer #1: Yes

Reviewer #2: Yes

3. Has the statistical analysis been performed appropriately and rigorously? 

Reviewer #1: Yes

Reviewer #2: Yes

4. Have the authors made all data underlying the findings in their manuscript fully available?

Reviewer #1: Yes

Reviewer #2: Yes

5. Is the manuscript presented in an intelligible fashion and written in standard English?

Reviewer #1: Yes

Reviewer #2: Yes

6. Review Comments to the Author

Reviewer #1: The author has answered the questions posed.

He has adopted the suggested advice.

He has added the necessary details.

I consider that this manuscript can now be published.

Reviewer #2: respected,

I looked carefully at the corrections you made. I also looked at the comments and corrections of other colleagues. I think that the work is now much better and ready for reception. I hope that the comments contributed to a better final version of the paper, and that is our common goal in our scientific community. Also, I hope we all learned something that can help us in our next works. good luck.

7. PLOS authors have the option to publish the peer review history of their article (what does this mean? ). If published, this will include your full peer review and any attached files.

**Do you want your identity to be public for this peer review?** For information about this choice, including consent withdrawal, please see our Privacy Policy .

Reviewer #1: **Yes: ** Bojan Masanovic

Reviewer #2: No

---

## [Author Response · Author response to Decision Letter 1]

17 Mar 2025

Response to Comment of Editor:

"The authors should revise the manuscript according to reviewers' suggestions. Additionally, the authors should improve the description of the concepts: substitution networks, substitution links, and further explain the meaning of the concept 'Higher density indicates closer substitution relationships among players, while lower density reflects looser connections.' The point is, what does 'closer substitution relationship' mean?"

Response:We greatly appreciate the editor’s insightful comments. To improve clarity and provide a deeper understanding of these key concepts, we have made the following revisions:

Substitution Networks: We have clarified that substitution networks is a social network constructed based on the substitution patterns of a team during matches where nodes represent players (substitutes or players substituted off) and edges represent substitution relationships during the match. These relationships capture the dynamics of player changes, which reflect substitution strategies.

Substitution Links: We further define substitution links as the connections between the player substituted off and the player who replaces them. We also explain how these links are formed and analyzed in our model.

Density of Substitution Networks: We have expanded on the explanation of network density. A higher density indicates that players on the team have a diverse set of substitution interactions, showing that the team’s substitution decisions involve a wider range of player combinations. A lower density, on the other hand, suggests that substitution decisions tend to involve only a small set of players, reflecting less variability in the team’s substitution strategy.

These revisions are included in Section Methods of the manuscript, where we now provide a more comprehensive explanation of these terms to ensure the concepts are fully understood by readers.

Response to Comments of Reviewer #1

We sincerely appreciate your positive feedback and recognition of our improvements. Your comments have been instrumental in enhancing the clarity and quality of our manuscript.

Response to Comments of Reviewer #2

Thank you very much for your encouraging words and constructive feedback. We are grateful for your support and recognition of our efforts to improve the manuscript.

---

## [Editor Report · Decision Letter 2]

19 Mar 2025

A Study on Substitution Characteristics and Competitive Performance of Chinese Super League Teams under the Five-Substitution Rule

PONE-D-25-03456R2

Dear Dr. Chen,

We’re pleased to inform you that your manuscript has been judged scientifically suitable for publication and will be formally accepted for publication once it meets all outstanding technical requirements.

Kind regards,

Bruno Travassos

Academic Editor

PLOS ONE
---

## [Editor Report · Acceptance letter]

PONE-D-25-03456R2

PLOS ONE

Dear Dr. Chen,

I'm pleased to inform you that your manuscript has been deemed suitable for publication in PLOS ONE. Congratulations! Your manuscript is now being handed over to our production team.

Kind regards,

on behalf of

Dr. Bruno Travassos

%CORR_ED_EDITOR_ROLE%

PLOS ONE